# ZWA: Viral genome assembly and characterization hindrances from virus-host chimeric reads; a refining approach

**Nikolas Dovrolis** \*, **Katerina Kassela, Konstantinos Konstantinidis,**
**Adamantia Kouvela, Stavroula Veletza, Ioannis Karakasiliotis** \*

Laboratory of Biology, Department of Medicine, Democritus University of Thrace, Alexandroupolis, Greece

\* ndovroli@med.duth.gr (ND); ioakarak@med.duth.gr (IK)

## Abstract

Viral metagenomics, also known as virome studies, have yielded an unprecedented number of novel sequences, essential in recognizing and characterizing the etiological agent and the origin of emerging infectious diseases. Several tools and pipelines have been developed, to date, for the identification and assembly of viral genomes. Assembly pipelines often result in viral genomes contaminated with host genetic material, some of which are currently deposited into public databases. In the current report, we present a group of deposited sequences that encompass ribosomal RNA (rRNA) contamination. We highlight the detrimental role of chimeric next generation sequencing reads, between host rRNA sequences and viral sequences, in virus genome assembly and we present the hindrances these reads may pose to current methodologies. We have further developed a refining pipeline, the Zero Waste Algorithm (ZWA) that assists in the assembly of low abundance viral genomes. ZWA performs context-depended trimming of chimeric reads, precisely removing their rRNA moiety. These, otherwise discarded, reads were fed to the assembly pipeline and assisted in the construction of larger and cleaner contigs making a substantial impact on current assembly methodologies. ZWA pipeline may significantly enhance virus genome assembly from low abundance samples and virus metagenomics approaches in which a small number of reads determine genome quality and integrity.

**Data Availability Statement:** The data underlying this article are available in BioSample repository with the following accession STUDY: PRJNA681030, SAMPLE: SAMN17358770,

## Author summary

For years now the study of viruses and their genetic composition has been important in their identification and classification. Especially in these times of the pandemic turmoil, accurate knowledge of a virus' exact genetic composition can help identify its strengths and weaknesses allowing us to track its evolution and assist in the development of vaccines and antiviral agents. The reconstruction of these genomic sequences is called the assembly process, a bioinformatics approach which can be complicated and full of pitfalls. This work identifies one such issue, concerning artifacts introduced in viral genomes from the new technologies of nucleic acid sequencing. The proposed algorithm helps alleviate this problem by tentatively removing these problematic regions while keeping the vast

EXPERIMENT: SRX9862216, RUN: sacha.fastq.gz (SRR13449040 - https://www.ncbi.nlm.nih.gov/sra/SRR13449040/). All the associated scripts can be found at https://github.com/ndovro/ZWA and https://hub.docker.com/repository/docker/ndovroli/zwa/.

**Funding:** All authors are co-financed by the European Union and Greek national funds through the Operational Program Competitiveness, Entrepreneurship and Innovation, under the call RESEARCH/CREATE/INNOVATE (project code: T1EDK/5000). The funders had no role in study design, data collection and analysis, decision to publish, or preparation of the manuscript.

**Competing interests:** The authors have declared that no competing interests exist.

majority of the genetic information required to produce a more complete viral genome. This work is anticipated to assist in the submission of higher integrity and accuracy viral genomes in public databases used for novel virus identification and characterization.

This is a *PLOS Computational Biology* Methods paper.

## Introduction

Identification of viruses through next generation sequencing (NGS) relies on the use of well curated viral databases [1]. Recent advances and the broad application of viral genome assemblies, from unbiased virus screening, have yielded an unprecedented amount of novel complete or partial genomes [2]. To date, NGS has been essential in the discovery of novel viruses and the analysis of complex samples in which multiple viruses are present. RNA-seq analysis and virus genome assembly from biological and environmental materials, containing novel human and animal viruses such as SARS-cov2, has greatly assisted their prompt and unbiased characterization [3]. Full genome assembly of SARS-cov2 was crucial for understanding the virus' biology and its origin along with the design of efficient molecular diagnosis methodologies and the molecular tracing of the pandemic [4]. In parallel, NGS analysis of complex clinical or environmental samples has introduced the "virome" [5–7], a new concept of the metagenomics era.

As viruses are a highly divergent form of life with multiple evolutionary origins [8,9], metagenomics approaches rely on unbiased DNA/RNA sequencing rather than targeted PCR based sequencing [10]. Alignment of the yielded DNA/RNA sequencing reads or further processed assemblies against virus databases are critical for the identification of viral components in a sample. Curated databases contain a very small portion of the currently known viral sequences. Open databases, such as GenBank, while they contain most of the current sequences, are poorly curated regarding the integrity and the accuracy of the submitted sequences. Indeed, various reports have highlighted contamination of sequences with bacterial moieties that were erroneously incorporated in the final assembly [11–13]. Such errors are especially important as they result in false positive identification of viruses that happen to contain in their proposed viral genomes parts of host DNA or RNA.

*De novo* assembly of novel viruses is based on a few thousand reads, or less, since environmental and clinical samples are usually not enriched [14]. Artifacts may arise, in part due simultaneous presence of multiple forms of the same RNA; positive and negative sense genomic RNAs or mRNAs, splicing variants and the presence of chimeric reads. Chimeric reads originate from chimeric DNA fragments during library preparation and have, in the past, been proven as a significant impediment in *de novo* transcriptome assemblies [15,16]. This is especially important during viral genome assembly from samples poor in viral sequences, where the complete removal of chimeric reads may result in partial genomes.

Contaminant sequences from viruses, bacteria and fungi have been an issue in DNA and RNA sequencing for years with a variety of approaches identifying and filtering them [17–20]. Previous works have highlighted the impact of metagenomic contaminants in genome or transcriptome assemblies and the role of NGS library preparations in the generation of contaminant reads that encompass exogenous nucleic acids [21]. In 2017, Dittami et al [12] recognizing the presence of contaminants in genomes, designed a bioinformatics tool which

analyzed published *Saccharina japonica* genomes and detected the contaminants proposing their tool as a part of standards bioinformatic analysis for the specific kelp. This tool focused on a breakdown of the genome into smaller contigs and the usage of BLASTn [22] to identify their origin. On the same year Fierst et al. [23] proposed a Machine Learning approach which, using nucleotide sequence similarities (adapted from microbial research techniques), filtered reads which were detected as contaminants, prior to final genome assembly. On the same principle, a variety of tools have been designed to identify and remove host-derived reads from metagenomic samples to assist in genome assembly. In fact, all current approaches and applied pipelines detect and curate genomes and transcriptomes post assembly or by excluding contaminating (chimeric) reads from the assembly. GenCoF [24], RNA-QC-chain [25] and FastqPuri [26], for example, are designed specifically as rRNA removal pipelines which implement a variety of tools for quality-control of sequences (based on sequencing/mapping quality) and complete removal of rRNA reads from the samples. The authors of MCSC [27] have also recognized the issue of contaminated assemblies and have developed a hierarchical cluster algorithm which distinguishes reads as contaminants and then tries to align and remove them. rRNAFilter [28], as the name suggests, is also a removal tool for rRNA reads with the added benefit of being alignment-free and based on read *k-mers* for the identification of rRNA. Fastq Screen [29] and SortMeRNA [30] adopt a similar approach to each other which involves mapping reads onto multiple genomes in order to identify and separate contaminant sequences. These latter approaches are often used in metagenomic studies, which aim to keep the rRNA reads in order to identify micro-organisms and discard host reads. For example during the identification of the new SARS-CoV-2 such an approach was utilized [31]. Testament to the fact that rRNA contamination remains a problem to this day, is the current development of new tools to leverage the issue.

In the present study, we sought to assess the impact of host-virus chimeras in the integrity of GenBank database and the role of rRNA-virus chimeric reads in the emergence of rRNA-contaminated virus genome contigs during *de novo* assembly. Furthermore, we developed an algorithm that enhanced the *de novo* assembly of viral genomes by upcycling chimeric reads, discarded by current pipelines.

## Materials and methods

### Reference databases

Two reference databases were developed based on sequences found on NCBI's Nucleotide database (NUCCORE). The first one, the Ribosomal Database (RiDB) contained 28S and 18S ribosomal RNA (rRNA) sequences from selected species that have been extensively studied in the past for the identification and assembly of viruses. For *Gallus gallus* (XR_003078040.1 and AF173612.1), for *Bos Taurus* (NR_036644.1 and NR_036642.1), for *Ixodes simplex* (KY457499.1 and KY457499.1), for *Anopheles sacharovi* (L22060.1 and X57172.1), for *Canis lupus* (XR_003129431.1 and XR_003129430.1), for *Ovis aries* (XR_003587871.1 and KY129860.1), for *Mus musculus* (NR_003279.1 and NR_003278.3) and for *Homo sapiens* (M11167.1 and X03205.1). The second database, the Viral Database (ViDB), was constructed using NCBI's e-utilities [32] with the keyword "viruses [ORGN]" and contains 3,197,815 virus sequence entries (on November 27 2019).

### Ribosomal contamination detection and identification

The RiDB was used to create, through a custom Python script (https://github.com/ndovro/ZWA/blob/main/mockup.py), mock fastq files that contained 4 million random "reads" (90-150bp) of Phred 30 quality for each of the host rRNA. These synthetic datasets were validated by aligning

them on the original sequences, in order to assess even distribution of reads, high depth and full coverage (**S1 Fig**). These mock files simulate sequencing runs and were aligned on the ViDB using 3 different aligners after creating the appropriate indices: BBMAP v 38.50 [33], STAR v. 020201 [34] and BOWTIE2 v. 2.3.4.1 [35]. All aligning runs were configured to keep reads that mapped on various contigs of the reference (different viral sequences of ViDB) and to ignore splicing events. This way we were able to identify sequences in Genbank that contain these chimeric rRNA-virus assemblies. The percentile (relative position) on each ViDB sequence where the rRNA contamination was detected was calculated and visualized all using Graphpad Prism 8 [36], Venny v.2.0.1 [37] and the R statistical programming language [38].

## RNA-seq

A pool of five *Anopheles sacharovi* individuals were homogenized and total RNA was extracted by TRIzol reagent (Thermo Fischer Scientific) according to the manufacturer's protocol. Whole transcriptome libraries were prepared from 500 ng of RNA extract, using the Ion Total RNA-Seq v2 Core Kit (#4479789, ThermoFisher Scientific) according to the manufacturer's instruction. In brief, the RNA library preparation involved RNA fragmentation, adapter ligation, reverse transcription and 14 cycles of PCR amplification using Ion Xpress RNA-Seq Barcode 1–16 Kit (#4475485, ThermoFisher Scientific). Quantification of the library was performed using Qubit Fluorometer high-sensitivity kit (ThermoFisher Scientific) and its median size was determined in LabChip GX Touch 24 (PerkinElmer). The libraries were loaded onto an Ion 540 chip, using Ion Chef (Thermo Fisher Scientific) and sequencing was carried out on an Ion GeneStudio S5 sequencer (ThermoFisher Scientific). Ion GeneStudio S5 sequencer returns the reads already quality trimmed.

## Sequence assembly and annotation

Trinity v2.8.5 [39] was used to *de novo* assemble contigs from the RNA-seq reads. The contigs produced by the assembler were analyzed with batch BLASTX (organism: viruses [taxid:10239]) for the identification of sequences that corresponded to viral genomes. Contigs corresponded to the Xanthi rhabdovirus (MW520377). The above contigs were used as reference for mapping the mock read file of RiDB in order to identify contigs encompassing rRNA moieties. Bam files of selected rRNA "contaminated" contigs were visualized using the Integrative Genomics Viewer (IGV) v. 2.5.3 [40] to identify the chimeric reads.

In order to identify the percentage of chimeric reads that corresponded to both viral genomes and rRNAs we used two reference files. One contained the rRNA genes' sequences for *Anopheles sacharovi* (MT808434, MT808462) and the other containing the Xanthi rhabdovirus sequence. BBMAP was used to map reads on each of these references. Using samtools v.1.7 [41] we identified the reads which mapped on pairings (virus vs rRNA) of the reference files. Venny v.2.0.1 was used to create Venn diagrams.

## Contamination cleanup and assembly optimization

After identifying the existence of chimeric reads as a part of the sequencing process, we set out to create a method that computationally removed ribosomal contamination from RNAseq reads without having to discard the entire read. We constructed and implemented a pipeline, the Zero Waste Algorithm (https://github.com/ndovro/ZWA/), which performs context-based read trimming by identifying the exact bases on each read where the rRNA has been fused and cutting them out leaving us with a "cured" read.

ZWA implements the following steps using well established bioinformatics tools:

1.  The run is initialized by providing the raw RNAseq reads in fastq format and an rRNA reference file appropriate to our sample.

2.  BWA mem v. 0.7.17-r1188 [42] is used to separate the reads that align to the rRNA reference (ribo.fastq) and those that don't (other.fastq)

3.  Using samtools [41] we identify the reads that contain soft-clipping in the ribo.fastq file, convert it to fasta using BBmap's [33] reformat script and discard the pure ribosomal sequences using faSomeRecords (https://github.com/santiagosnchez/faSomeRecords).

4.  On the soft-clipped sequences from the previous step we perform a local BLASTn [22] v.2.9.0 query versus the ribosomal reference to locate the start and end bases of their chimeric part.

5.  In this step we excise the ribosomal part from the sequences.

6.  Finally, we combine the reads that did not align to the ribosomal reference in step 2 with the "cured" sequences of step 5 and perform a Trinity [39] assembly on the composite fasta file.

The concept behind this pipeline is depicted in **Fig 1.** Additional information regarding the implementation of this pipeline is stated both on the github page and as comments in the ZWA script. In addition, a Docker image based on Ubuntu 20.04 has been created containing all the appropriate prerequisite tools and the ZWA script. It can be found at https://hub. docker.com/repository/docker/ndovroli/zwa/ and can be deployed on an available server or cloud, as is, maximizing performance.

## Results

Building a pipeline for the unbiased identification of viruses from various species using the GenBank database we often encountered recurrent hits of specific entries regardless of the type or origin of the specimen. Using Basic Local Alignment Search Tool (BLAST) [22] these entries were found to encompass host RNA sequences, the vast majority of which were annotated as host ribosomal RNAs.

### Chimeric virus-rRNA entries in GenBank

Screening GenBank for chimeric assemblies that encompassed rRNA stretches derived from various host organisms we identified 38 erroneous entries. These 38 entries were matched using three different aligners with either 28S or 18S sequences in our Ribosomal database (RiDB), constructed using the rRNAs from medically or veterinary important animal species. BBMAP (38 hits) and STAR (32 hits) appeared to be the most sensitive, while, BOWTIE2 was less efficient (9 hits) (**Fig 2**). All the above sequences were manually (BLASTn [22]) validated to encompass an rRNA stretch. These results appear to be independent of the sequencing platform used.

As chimeric genomes may arise due to the presence of chimeric reads during assembly it was hypothesized that these contaminating sequences would be the cause of premature assembly termination, and thus they were expected to map at the extremities of these "contaminated" genomes. In order to assess the position of rRNA contaminants we mapped their relative position on the sequence of the respective Genbank virus entry. As expected, the majority of the rRNA sequences were located at the extremities while, to our surprise, a significant portion was located in the middle of the viral sequences (**Fig 3**). **S1 Table** also provides a complete list of these sequences along with the starting position and length of contamination for future reference.

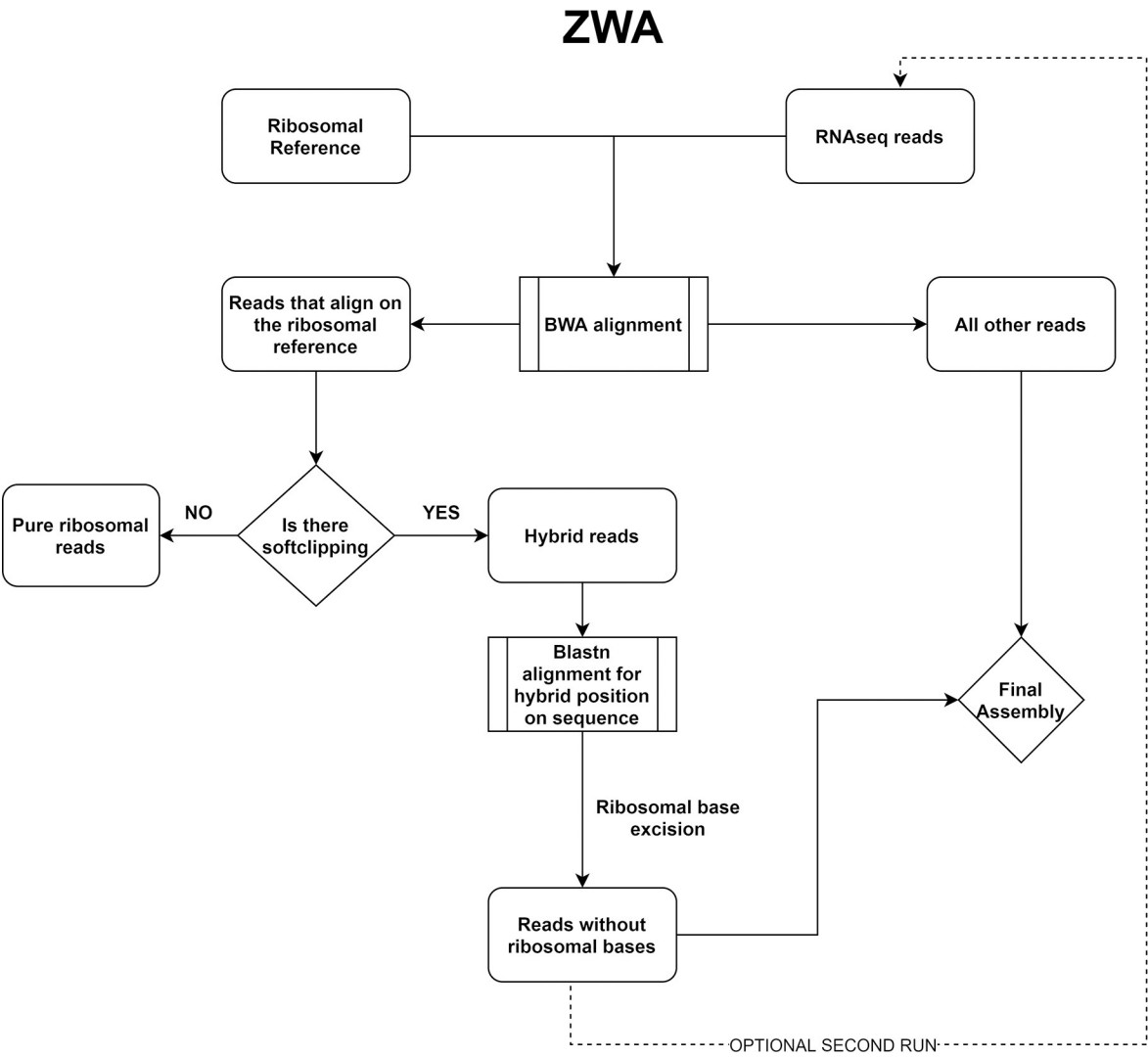

**Fig 1. Flowchart of the chimeric reads clean-up pipeline.**

## Chimeric reads as inherent problem of RNA-seq

In order to analyze the impact of chimeric reads in the *de novo* assembly of RNA viruses we used the mosquito *Anopheles sacharovi*, a known vector of human pathogens, as a paradigm. In a pool of such mosquitoes (environmental samples) we performed RNA-seq analysis of total RNA. *De novo* assembly using TRINITY resulted in a number of contigs that corresponded to viral sequences, as identified by batch BLASTx [22] analysis. The virus contigs identified, corresponded to Xanthi rhabdovirus genome (MW520377). The vast majority of the viral sequences encompassed at least one (up to 4) moiety corresponding to ribosomal RNA, usually at the extremities of the contigs. Analysis of individual reads that had passed all quality controls and mapped on chimeric contigs, revealed that chimeric reads served as bridges in the assembly of chimeric contigs (**Fig 4**). These reads were singletons and were only presented multiple times as PCR duplicates (same sequence and size products). Either in the extremities or in the middle of the contigs and even if such chimeric reads were a small fraction of the total mapped reads on any particular chimeric contig, they were detrimental to the

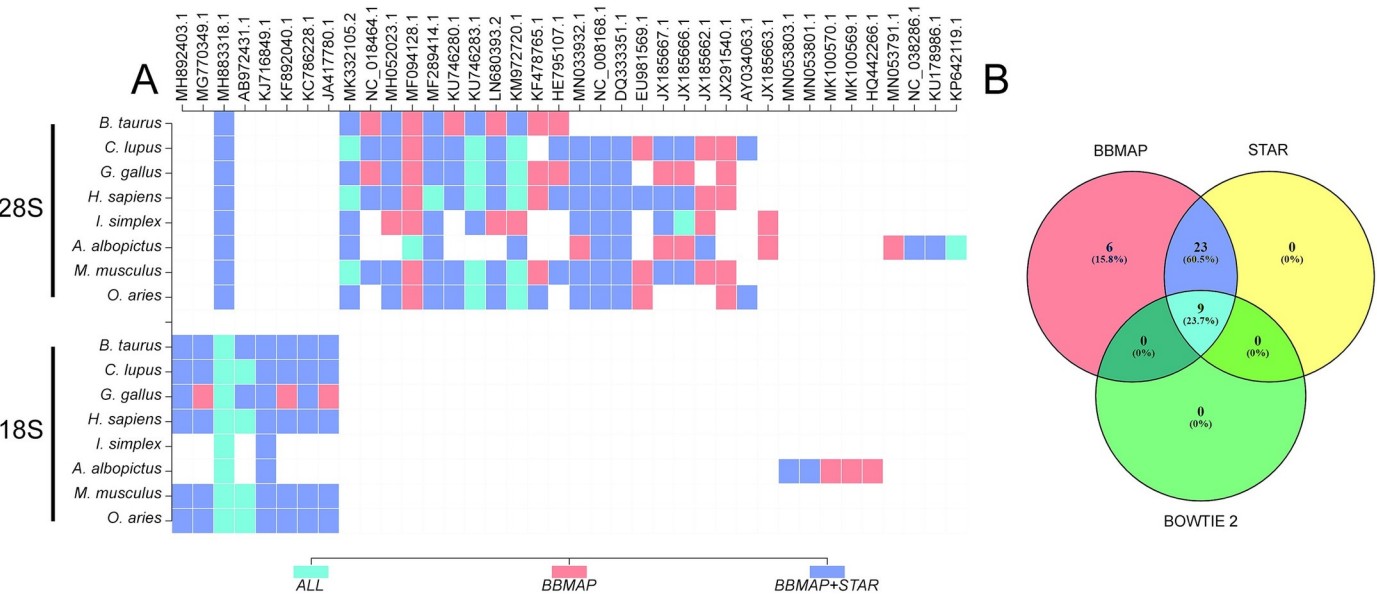

**Fig 2.** A) Match matrix of ribosomal hits on viral sequences. X-axis contains all the sequences from the ViDB that are chimeric and Y-axis shows the 28S and 18S rRNA sequences from our RiDB. Colours represent the aligners used to identify these chimeric (other combinations than those depicted resulted in null). B) Venn diagram of the chimeric hits identified by the 3 aligners for all host species.

achievement of a correct assembly. One such chimeric contig is presented in **Fig 4** depicting the effect of chimeric reads on the emergence of such a contig. The stochastic methodology of the assemblers resulted in the emergence of various such chimeric solutions in every test of the pipeline.

To identify the percentage of chimeric reads between rRNAs and a viral genome in an NGS reaction we aligned total raw reads against *Anopheles sacharovi* rRNAs and Xanthi rhabdovirus. A significant proportion of the reads that mapped to the viral genome mapped also to the rRNAs. From a total of ~3.5 million reads (quality trimmed), Xanthi rhabdovirus was represented by 1434 reads of which 836 (58%) were chimeric according to SortMeRNA or 807 (56%) according to BWA (**Fig 5**).

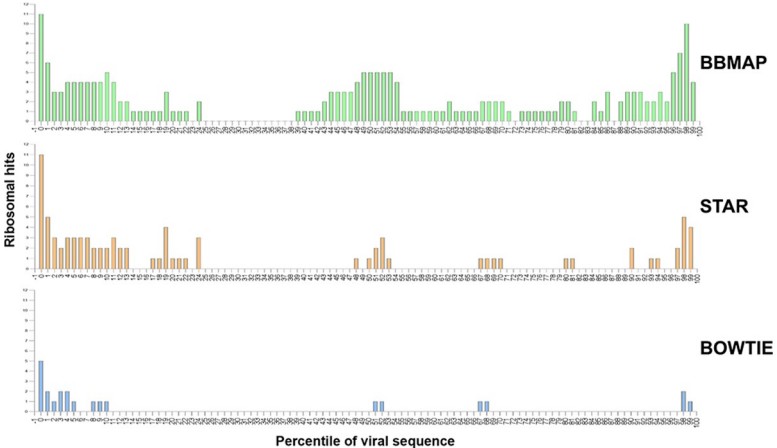

**Fig 3. Alignment of sequences from the RiDB on the ViDB by all three aligners.** Bar chart representation of the number of rRNA hits on viral sequence GenBank entries per percentile of the sequence. X-axis represents the percentile of viral sequence in which the ribosomal sequences start.

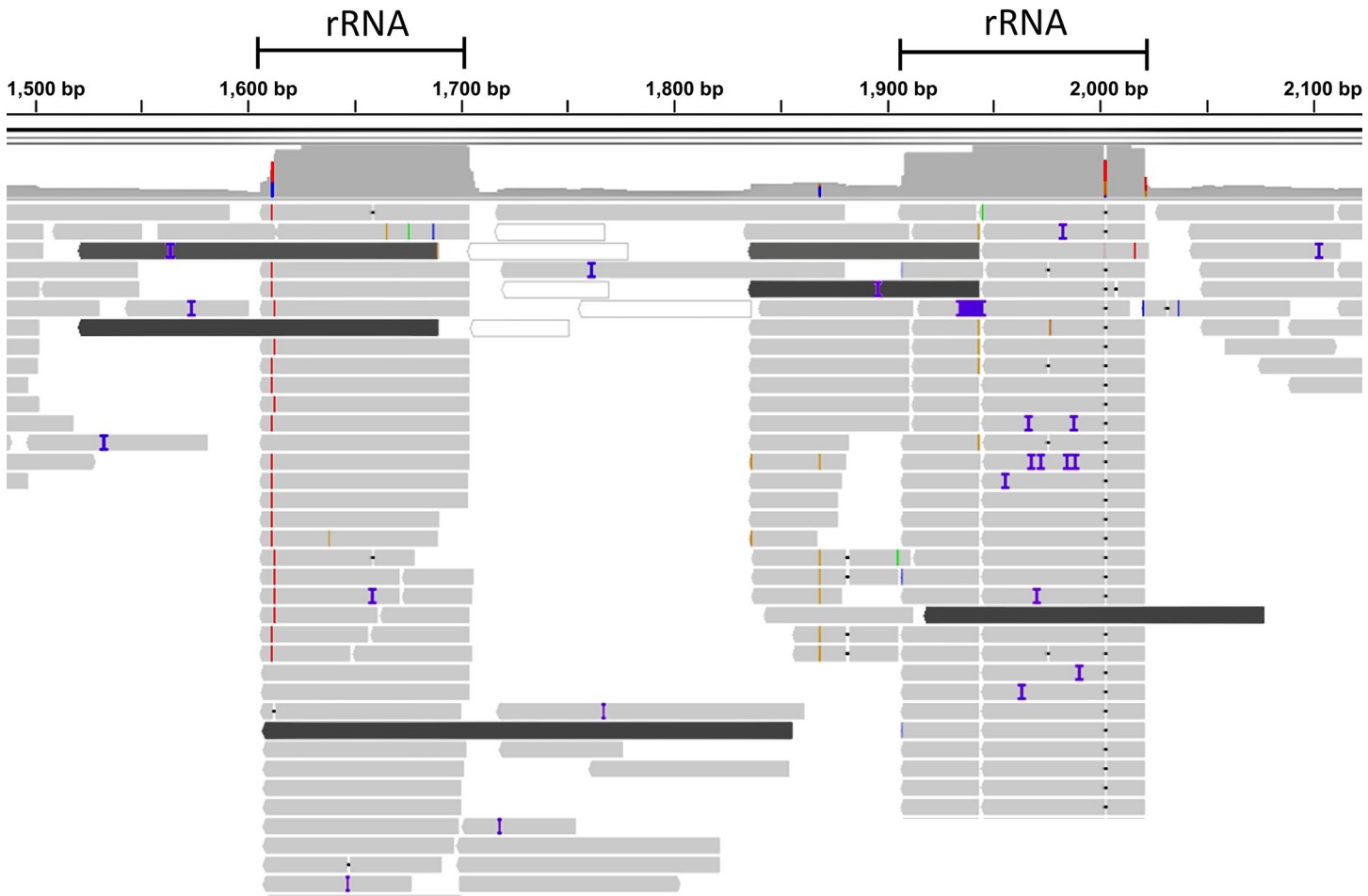

**Fig 4. Example of visualization, in Integrative Genomics Viewer, of sequencing reads aligned on rRNA/viral chimeric contigs generated via de-novo assembly.** Dark grey reads indicate chimeric that serve as bridges in the assembly between rRNA and viral sequences. Coloured lines represent gaps and mutations against reference sequence. Reads have been reduced for illustration purposes.

### Improving assembly of viruses through specific trimming of chimeric reads

*De-novo* assembly of viral genomes directly from raw reads of total-RNA seq very often results in the chimeric contigs such as the one presented in **Fig 4** that often terminate prematurely the alignment, resulting in both contaminated and aberrant contigs. Alignment of all the contigs that derived from direct *de-novo* assembly on the Xanthi rhabdovirus genome revealed the extent of aberrancy and contamination of these contigs. Although the contigs cover a large extent of the viral genome, the incorporation of rRNA contamination is often unavoidable as the vast majority of the contigs are flanked by rRNA moieties (**Fig 6,** colored lines). The reduced mean contig length (804.3 bp) and the reduced mean coverage of the contigs (84.00%, percent of the contig that maps to the virus genome) were indicative of the detrimental effect of the contaminants in virus genome assembly (**Table 1**).

Frequent incorporation of ribosomal stretches in the *de novo* assemblies from total RNA-seq indicated the need of a pipeline that improves assemblies by removing rRNA contaminations. Current methodologies using read-filtering algorithms such as SortMeRNA resulted in the significant reduction of reads that map to the viral genome (598 reads from 1434 total) (**Table 1**). RRNA-virus chimeric reads were binned with the rest of the rRNA-containing

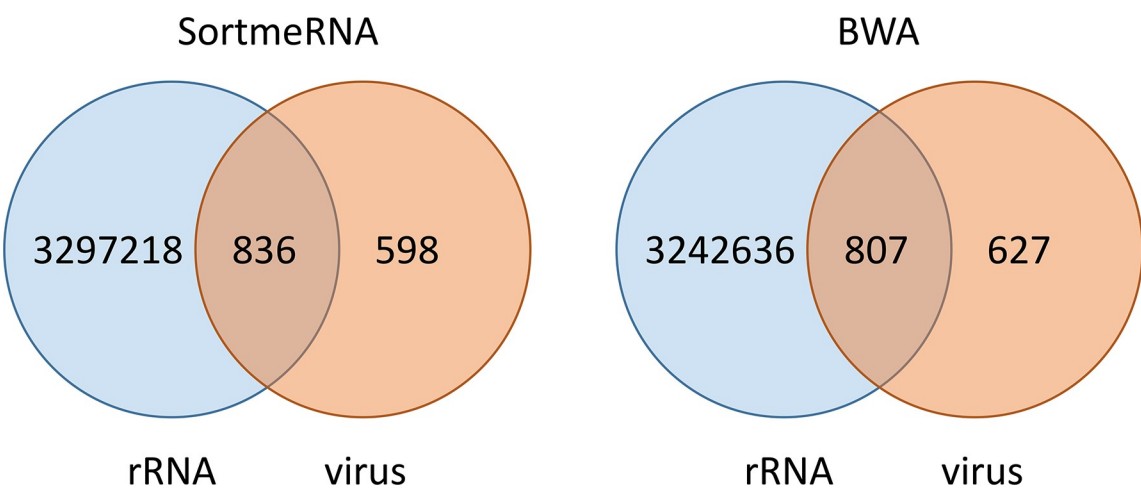

**Fig 5. Venn diagrams highlighting the chimeric rRNA/viral reads produced by the NGS process on Xanthi rhabdovirus from Anopheles sacharovi sample.**

reads and were not used during *de-novo* assembly. On the other hand a significant number (836 using SortMeRNA and 807 using BWA) of rRNA-containing reads could partially align (non-rRNA moiety) on the viral genome signifying a great loss of genetic information. Such a loss of information was detrimental in the assembly, as although it assisted in less incorporation of rRNA moieties (89% mean coverage of contigs) the mean contig length was not improved (**Table 1**).

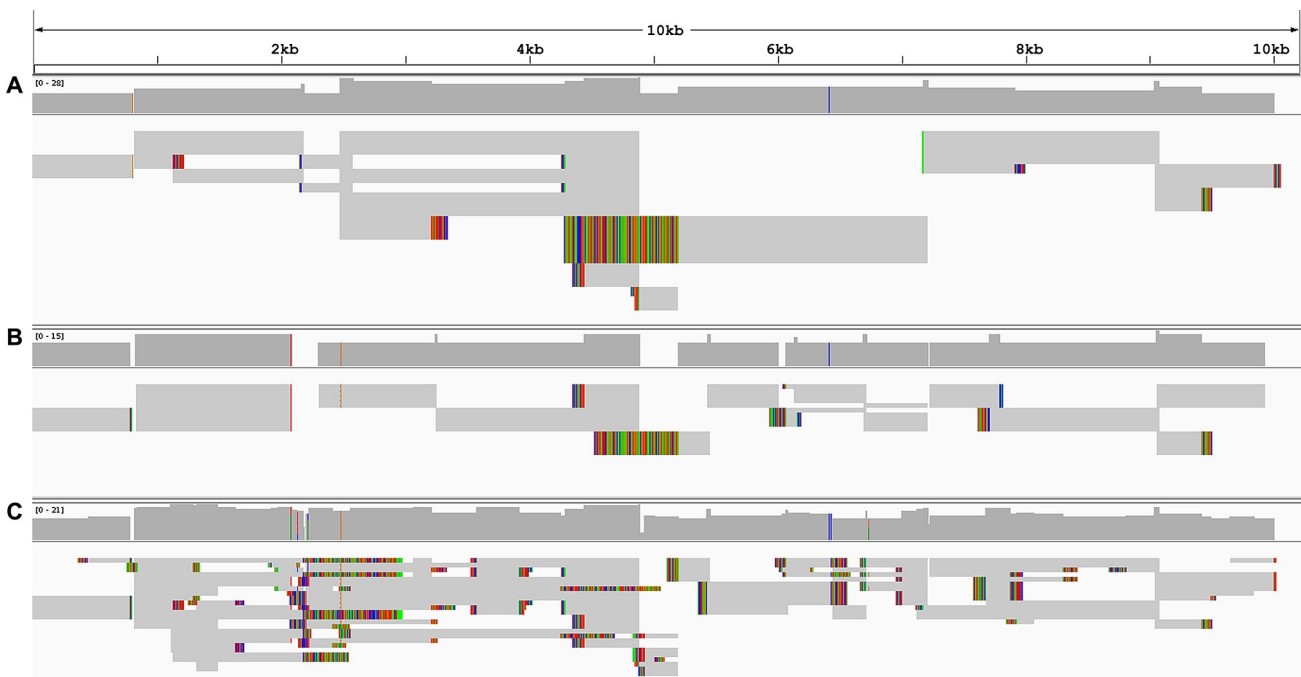

**Fig 6.** IGV representation of chimeric contigs between rRNA and viral sequences mapped on the Xanthi rhabdovirus genome after A) context-based trimming of reads for rRNA contaminants, B) SortMeRNA rRNA reads filtering and C) raw data assembly. Coloured lines are softclips of the contigs that do not align to the viral genome and represent chimeric moieties of the contigs.

**Table 1. Characteristics of the assembled contigs after ZWA, SortMeRNA and without treatment (raw reads).** Mean coverage, genome coverage and the respective used reads after mapping on virus genome (± standard deviation), *n = 5 Trinity assemblies **5 Trinity assemblies.

| | MEAN CONTIG LENGTH* (bases) | MAPPED ON VIRUS GENOME | | |
|---|---|---|---|---|
| | | MEAN COVERAGE OF CONTIGS* (%) | VIRUS GENOME COVERAGE** (%) | USED READS FOR ASSEMBLY** |
| Assembly after ZWA | 1209.4 ±43 | 91.06 ±0.44 | 97.97 | 1335 |
| Assembly after SortMeRNA | 793.7 ±25 | 89.00 ±1.49 | 90.62 | 598 |
| Assembly using raw reads | 804.3 ±118 | 84.00 ±3.04 | 97.61 | 1434 |

In order to make chimeric reads useful for an efficient *de novo* assembly without risking the introduction of contaminants we developed a novel pipeline; Zero Waste Algorithm, ZWA. Through this pipeline the rRNA portion of rRNA reads that partially (<90%) mapped to the rRNA database (RiDB) was context-base trimmed from these chimeric reads, leaving the non-rRNA portion for use in the assembly pipeline. During this read-cleaning process, 656 reads out of the 807 chimeric reads (BWA process) were cured from their rRNA moiety (**Fig 7**). However, as chimeric reads are more complicated and may incorporate more than one rRNA moiety the pipeline was run for a second round of read-cleaning during which 52 reads out of the 656 cured in the previous run were further cured from an additional rRNA moiety (**Fig 7**). Merging the reads that were cured through both rounds of ZWA (cured reads) with the non-rRNA reads during the initial sorting (BWA process) were able to use twice as many reads (1335) as those sorted by SortMeRNA during virus genome assembly (**Table 1**).

## Discussion

Identification of novel viruses in clinical and environmental specimens usually employs pipelines that match RNA-seq *de-novo* assemblies to databases that contain viral sequences such as the NCBI database GenBank. However, the ever-increasing rate of submissions in Genbank and the lack of several steps of sequence integrity validation have resulted into sequence artifacts that interfere with processes such as sequence alignments and contig annotation. In an effort to curate the Genbank database ~8000 viral sequences from the 3,400,000 available viral genomes were included in the RefSeq database [43]. More specialized databases include NCBI

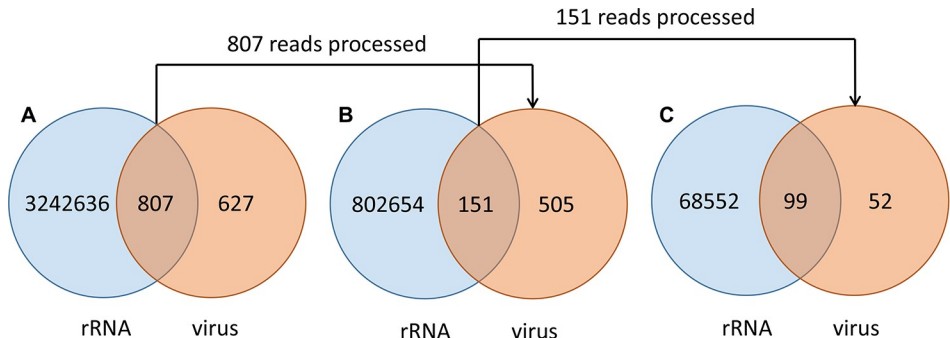

**Fig 7. Ribosomal RNA read cleaning process outcome.** Venn diagrams depicting (A) the initial sorting of rRNA reads and non-rRNA reads with numbers referring to rRNA reads and virus reads in the two bins. The intersection are rRNA-virus chimeric reads. (B) The rRNA reads recognized as chimeric (<90% rRNA) and the resulted cured virus reads after ZWA. The intersection are the remaining rRNA-virus chimeric reads. (C) The rRNA reads recognized in the second round as chimeric (<90% rRNA) and the resulted cured virus reads after ZWA. The intersection are the discarded remaining rRNA-virus chimeric reads.

virus, EBI metagenomics, IMG/VR v.2.0, Metavir2 and iVirus. However, the vast majority of newly identified viral sequences or partial viral sequences, that are products of *de novo* assemblies through viral metagenomics screening of new hosts, are rarely included in curated databases. Alignments against GenBank are by far the most frequent pipeline when assessing viral metagenomics. Database assisted viral assembly through alignment to GenBank (like what the tool VrAP, Viral Assembly Pipeline does) [44] is also a novel methodology for more accurate virus assembly and identification that relies on GenBank integrity.

During the implementation of a pipeline for unbiased high-throughput identification of novel viruses, we came across GenBank entries that encompass ribosomal RNA moieties and came up as false positive identifications. Screening GenBank for such entries, we identified 38 submissions with significant ribosomal RNA contamination. Such contamination was mainly located at the extremities of the sequences although several entries encompassed the contamination in the middle. Chimeric assembly artifacts may mainly arise due to chimeric reads that emerge as library preparation byproducts. As the virus genome entries are increasing rapidly in sequence databases due to the wide application of viral metagenomics, there is an eminent need to avoid the generation of more contaminated entries from virome approaches. Viral genomes during virus metagenomics are often encountered in a small fraction of individuals or are diluted out in environmental samples. For example, genomes of human pathogens such as West-Nile and Ross River virus are present in less than 1% of vectors such as mosquitoes [45,46], while analysis of just 10 L of sea water in China yielded 4,593 nearly full-length RNA virus polymerase genes [47].

In order to refine the assembly process and reduce the generation of contaminated viral genomes in such approaches we assessed the effect of host rRNA-virus chimeric reads in the assembly of low abundance virus genomes. For this reason, in a working example we used a pool of *Anopheles sacharovi* mosquitoes, an important human pathogen vector [48] sequenced using total RNA-seq, for the identification of novel RNA virus genomes. During the analysis we noted that a significant fraction (half) of reads that mapped to Xanthi rhabdovirus genome, a virus previously identified in this species, contained also an rRNA moiety.

Previous attempts to minimize rRNA contamination during total RNA-seq included poly (A) selection and ribodepletion before library preparation or computational filtering of reads that mapped on ribosomal RNAs [24–30,49]. Poly(A) selection is prohibited during virus metagenomics as several virus families do not contain a poly(A) tail in their genomic RNA [50], while ribodepletion is not always efficient as it is species specific [51] and the organism or pool of organisms (environmental samples) during virome analysis may vary significantly. On the other hand, bioinformatics pipelines efficiently filter reads that map to the host rRNAs removing them from the assembly procedure. Such pipelines are part of current the state-of-the-art methodologies used for RNA virus genome assembly, while in others no rRNA filtering was performed prior to assembly [47,52,53]. Applying such a filtering procedure using SortMeRNA, a dedicated software for ribosomal RNA sorting, we observed that the algorithm would process rRNA-virus chimeric reads together with the rRNA reads, binning them accordingly. Although such a procedure generated cleaner contigs during assembly, the significant reduction of virus reads (half) had detrimental effect in obtaining important contigs for the assembly of the complete virus genome.

In an attempt to feed back the filtered out chimeric reads in the assembly pipeline we developed a novel pipeline that would cure rRNA-containing chimeric reads using a context-based trimming algorithm; as opposed to quality-based trimming. The ZWA algorithm, using a BLASTn-directed procedure, was able to remove accurately the rRNA moieties from the chimeric reads rendering them useful for the enhancement of the *de novo* assembly efficiency. This is contrary to other past algorithms which trim reads using a virus reference guided

approach that rely on existing virus databases, rendering such algorithms inefficient in virus metagenomics and the identification of novel virus genomes [54]. The ZWA algorithm returned the majority of the initially discarded reads, without their rRNA bases, into the assembly of the virus genome achieving higher genome coverage with bigger and cleaner contigs. The ZWA approach resulted overall in a smoother and less wasteful assembly procedure superior to current approaches that involve rRNA filtering algorithms.

A limitation of this procedure is the possibility to miss naturally occurring RNA virus recombinant genomes that have incorporated small fragments of ribosomal RNAs. Such a very rare event has been presented twice in the literature reporting a presumable incorporation of 54nt and 15nt fragments of ribosomal RNA (28S rRNA) in influenza virus [55] and poliovirus [56] genomes respectively. Since then, there has been no other similar report. Of note, these 2 genomes are not included in GenBank so our search algorithm did not identify them. However, assembly of such naturally occurring recombinant genomes should present a clear break in the recombination site after the application of the ZWA algorithm, as no other bridging read would be present in the sample.

## Conclusions

Growing needs to screen habitats for potential emerging viral pathogens combined with the increasing affordability and the reduced complexity of NGS procedures increase the importance of maintaining the integrity and reliability of virus databases. Our work highlights an aspect in the plethora of NGS artifacts and proposes a pipeline that significantly reduces contaminated and aberrant viral genome assemblies by implementing a precise upcycling of previously discarded chimeric reads.

## Supporting information

**S1 Table. GenBank Accession IDs of virus sequences containing ribosomal sub-sequences along with the starting position and length of the moieties.**
(DOCX)

**S1 Fig. Alignment of mock human 28S rRNA reads on the human 28S reference sequence (M11167.1) for validation.** X-axis shows the bp positions of the reference and Y-axis shows how many reads of the mock file were aligned on each position.
(DOCX)

## Author Contributions

**Conceptualization:** Ioannis Karakasiliotis.

**Data curation:** Nikolas Dovrolis.

**Formal analysis:** Nikolas Dovrolis.

**Funding acquisition:** Ioannis Karakasiliotis.

**Investigation:** Katerina Kassela, Konstantinos Konstantinidis, Adamantia Kouvela.

**Methodology:** Nikolas Dovrolis, Ioannis Karakasiliotis.

**Project administration:** Ioannis Karakasiliotis.

**Resources:** Stavroula Veletza.

**Software:** Nikolas Dovrolis.

**Supervision:** Ioannis Karakasiliotis.

**Visualization:** Nikolas Dovrolis, Ioannis Karakasiliotis.

**Writing – original draft:** Nikolas Dovrolis, Ioannis Karakasiliotis.

**Writing – review & editing:** Nikolas Dovrolis, Stavroula Veletza, Ioannis Karakasiliotis.

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
