## [Decision Letter · Decision Letter 0]

18 May 2021

Dear Dr. Dovrolis,

Thank you very much for submitting your manuscript "Viral genome assembly and characterization hindrances from virus-host chimeric reads; a refining approach" for consideration at PLOS Computational Biology.

As with all papers reviewed by the journal, your manuscript was reviewed by members of the editorial board and by several independent reviewers. In light of the reviews (below this email), we would like to invite the resubmission of a significantly-revised version that takes into account the reviewers' comments.

We cannot make any decision about publication until we have seen the revised manuscript and your response to the reviewers' comments. Your revised manuscript is also likely to be sent to reviewers for further evaluation.

Sincerely,

Christos A. Ouzounis

Associate Editor

PLOS Computational Biology

Ilya Ioshikhes

Deputy Editor

PLOS Computational Biology

Reviewer's Responses to Questions

**Comments to the Authors:**

Reviewer #1: The paper presents an improved approach for the viral genome assembly using virus-host chimeric reads but reducing artifacts.

First of all, I advise to improve the title to a new one: "METHOD-NAME: a refined (?) viral genome assembly approach by characterization hindrances from virus-host chimeric reads".

Next, I have a few major comments to be addressed:

1. The authors presented an evidence that a group of deposited sequences encompassed ribosomal RNA (rRNA) contamination in viral genomes after genome assembly. I would suggest to check not only ribosomal RNA, but whole human genome sequences, since the contamination may be done by the pieces of "junk DNA" and/or coding regions from human genome as well as other hosts. Please provide an analyzes per host and per all possible viruses by a categories: DNA viruses and RNA viruses. For the viruses' classification purposes the recent publication in the FEBS journal may be used: https://doi.org/10.1111/febs.15835

2. Now the question arises: how do you distinguish between the "contamination" by rRNA and the horizontal gene transfer (HGT) events? Please collected reported HGT cases, make a test set, and verify your method such that you will be able to separate between the artifacts and the real HGT events. Please make available the HGT cases for users as well as other test sets.

3. Please show the statistics on the observed chimeric cases for different hosts, not just for the human host, but also for all the hosts/viruses collected by the categories mentioned in 1. The "goal standard contigs" set should be produced such that any viral analyses may trim easily these previously observed chimeric reads in human/non-human hosts.

Reviewer #2: Please see the attached comments.

**Have the authors made all data and (if applicable) computational code underlying the findings in their manuscript fully available?**

Reviewer #1: **No: **Supplementary Data should be added with the test sets as mentioned in my review.

Reviewer #2: Yes

PLOS authors have the option to publish the peer review history of their article (what does this mean?). If published, this will include your full peer review and any attached files.

Reviewer #1: No

Reviewer #2: No
---

## [Decision Letter · Decision Letter 1]

6 Jul 2021

Dear Dr. Dovrolis,

Thank you very much for submitting your manuscript "ZWA: Viral genome assembly and characterization hindrances from virus-host chimeric reads; a refining approach" for consideration at PLOS Computational Biology. As with all papers reviewed by the journal, your manuscript was reviewed by members of the editorial board and by several independent reviewers. The reviewers appreciated the attention to an important topic. Based on the reviews, we are likely to accept this manuscript for publication, providing that you modify the manuscript according to the review recommendations.

Sincerely,

Christos A. Ouzounis

Associate Editor

PLOS Computational Biology

Ilya Ioshikhes

Deputy Editor

PLOS Computational Biology

[LINK]

Reviewer's Responses to Questions

**Comments to the Authors:**

Reviewer #1: The manuscript has been improved significantly as a result of the revision.

I suggest the following technical improvement:

please develop a more time-efficient pipeline possibly with the use of GPU- or cloud-computing to extend the presented proof-of-concept algorithm and provide the online availability for the scientific community as a link/availability issue in the manuscript.

Reviewer #2: Authors have responded to all the comments and modified the manuscript. I would suggest acceptance of the manuscript with the title ZWA: Viral genome assembly and characterization hindrances from virus-host chimeric reads; a refining approach.

**Have the authors made all data and (if applicable) computational code underlying the findings in their manuscript fully available?**

Reviewer #1: Yes

Reviewer #2: None

PLOS authors have the option to publish the peer review history of their article (what does this mean?). If published, this will include your full peer review and any attached files.

Reviewer #1: No

Reviewer #2: No

Figure Files:

Data Requirements:

Reproducibility:

References:

---

## [Decision Letter · Decision Letter 2]

24 Jul 2021

Dear Dr. Dovrolis,

We are pleased to inform you that your manuscript 'ZWA: Viral genome assembly and characterization hindrances from virus-host chimeric reads; a refining approach' has been provisionally accepted for publication in PLOS Computational Biology.

Best regards,

Christos A. Ouzounis

Associate Editor

PLOS Computational Biology

Ilya Ioshikhes

Deputy Editor

PLOS Computational Biology

Reviewer's Responses to Questions

**Comments to the Authors:**

Reviewer #1: I recommend to accept this paper.

Reviewer #2: no extra comments.

**Have the authors made all data and (if applicable) computational code underlying the findings in their manuscript fully available?**

Reviewer #1: Yes

Reviewer #2: Yes

PLOS authors have the option to publish the peer review history of their article (what does this mean?). If published, this will include your full peer review and any attached files.

Reviewer #1: No

Reviewer #2: No

---

## [Editor Report · Acceptance letter]

4 Aug 2021

PCOMPBIOL-D-21-00526R2 

ZWA: Viral genome assembly and characterization hindrances from virus-host chimeric reads; a refining approach

Dear Dr Dovrolis,

I am pleased to inform you that your manuscript has been formally accepted for publication in PLOS Computational Biology. Your manuscript is now with our production department and you will be notified of the publication date in due course.

With kind regards,

Melanie Wincott
